DATA RELEASE

# MODIRISK: Mosquito vectors of disease, collection, monitoring and longitudinal data from Belgium

Wim Van Bortel[1], Veerle Versteirt[2], Wouter Dekoninck[3], Thierry Hance[4], Dimitri Brosens[5],* and Guy Hendrickx[6]

1 Institute of Tropical Medicine (ITG), Nationalestraat, 155, 2000, Antwerpen, Belgium
2 Agency for Nature and Forests, (ANB), Havenlaan 88 b75, 1000, Brussels, Belgium
3 Royal Belgian Institute for Natural Sciences (RBINS), Vautierstraat 29, 1000, Brussels, Belgium
4 Université Catholique de Louvain, Croix du sud 4-5, 1348 Louvain-la-Neuve, Belgium
5 Research Institute for Nature and Forest (INBO), Havenlaan 88 b73, 1000, Brussels, Belgium
6 Avia-GIS NV, Risschotlei 33, 2980, Zoersel, Belgium

## ABSTRACT

The MODIRISK project studied mosquito biodiversity and monitored and predicted biodiversity changes, to actively prepare to address issues of biodiversity change, especially invasive species and new pathogen risks. This work is essential given continuing global changes that may create suitable conditions for invasive species spread and the (re-)emergence of vector-borne diseases in Europe. Key strengths of MODIRISK, in the context of sustainable development, were the links between biodiversity and health and the environment, and its contribution to the development of tools for describing the spatial distribution of mosquito biodiversity. MODIRISK addressed key topics of the global Diversitas initiative, which was a main driver of the Belspo 'Science for a Sustainable Development' research program. Three different MODIRISK datasets were published in the Global Biodiversity Information Facility (GBIF): the Collection dataset (the Culicidae collection of the Museum of Natural History in Brussels); the Inventory dataset (data from the MODIRISK inventory effort); and the Longitudinal dataset (experiment data used for risk assessments).

Submitted: 28 February 2022

* Corresponding author. E-mail: dimitri.brosens@inbo.be

Preprint submitted at https://doi.org/10.5281/zenodo.6334125

Included in the series: *Vectors of human disease* (https://doi.org/10.46471/GIGABYTE_SERIES_0002)

**Subjects** Ecology, Biodiversity, Taxonomy

## DATA DESCRIPTION

Mosquito-borne diseases are prime candidates as (re-)emerging vector-borne diseases in Europe. Knowledge of the taxonomic and functional biodiversity of both endemic and invasive mosquito species, as well as the factors driving changes, was missing in Belgium. Acquiring this knowledge is an essential step towards understanding the current risk and preparing action plans for future threats.

## CONTEXT

The MODIRISK project was established with objectives to:

(1) Create an inventory of endemic and invasive mosquito species in Belgium, considering environmental and taxonomic elements of biodiversity (the 'Collection dataset')

(2) Assess the population dynamics of selected endemic and invasive mosquito species and their interrelationships (the 'Longitudinal dataset')

**Figure 1.** Overview of the MODIRISK data. Map of Belgium with red dots showing collection locations of Culicidae in the RBINS collection (the 'Collection dataset'); blue dots represent Inventory collection locations (the 'Inventory dataset'); and black dots represent locations of collections in the 'Longitudinal study' dataset.

(3) Model mosquito biodiversity distribution at a 1-km resolution, (the 'Inventory')

(4) Disseminate project outputs.

The three datasets, which originated during the MODIRISK project (Figure 1), were standardized to Darwin Core [1] and published by the Royal Belgian Institute for Natural Sciences (RBINS) through the Integrated Publishing Toolkit (IPT) [2] of the Belgian node of the Global Biodiversity Information Facility (GBIF).

## METHODS

During the first phase of the project (2007–2008), the focus was on inventory activities; setting up laboratory experiments to study life history traits of *Culex pipiens* in relation to temperature, and the first selection of models based on the field results. During the second phase of the project (2009–2010), the focus was on spatial model building and validation, the longitudinal study, and the dynamics of selected indigenous and invasive species found during the first-phase inventory and on more population genetics-driven research.

Thus, three different datasets were developed during the MODIRISK project: the 'Collection' dataset [3, 4], dealing with both historic and recent Culicidae specimens, the 'Inventory' dataset [5], dealing with the current inventory of Culicidae in Belgium, and the

'Longitudinal study' dataset [6], dealing with questions about risk assessment, outbreaks, and possible distribution.

These three datasets are closely linked, but are published as three different Darwin Core archives. The Inventory dataset 'MODIRISK: Monitoring of Mosquito Vectors of Disease (Inventory)' was first published in 2013 (occurrence core), while the Collection dataset (occurrence core) and the longitudinal study (event core) were published in 2017.

The project was coordinated by the Institute of Tropical Medicine, Antwerp, Belgium.

### The Collection dataset [3, 4]

In the early 1900s, Maurice Goetghebuer and Michel Bequaert collected many mosquitos from all over Belgium and established on of the most representative and richest collections of Belgian Diptera. These are preserved at RBINS [7]. The RBINS Culicidae collection comprises four parts: a general Belgian collection, the Goetghebeur subcollection, the Becquart subcollection, and a subcollection of unidentified specimens known as the 'supplements' (Figure 2). The Bequaert subcollection was mainly collected between 1912 and 1958, and comprises 135 voucher specimens. The Goetghebuer subcollection was collected between 1909 and 1946 (mainly between the period 1910–1930), and comprises 268 specimens. In the general Belgian collection, there are 241 specimens present, all collected between 1878 and 1967 (mainly between 1880 and 1925). The supplements comprise the largest subcollection, with 737 specimens collected between 1892 and 2005 (mainly during 1920–1960).

All 1381 specimens (24 species) in the RBINS collections were re-identified and digitized during the MODIRISK project. Furthermore, all voucher specimens from the available collections were re-identified at the species level [8]. Seventy seven percent of the specimens were collected between 1910 and 1960, with most specimens collected between 1940 and 1950. The intensity of research and mosquito-sampling fluctuated during this period, as revealed by the number of voucher specimens per decade (Figure 3). The oldest specimens (collected in 1878) are deposited in the general Belgian collection. In this collection, 16 species were discovered; in the Bequaert, Goetghebuer and in the supplements collections, respectively, 18, 21 and 20 species were counted. *Culex pipiens* and *Culiseta annulata* were the most abundant recorded species present in the collection (Figure 4), as were many voucher specimens of *Aedes punctor*. 1374 records were published on the GBIF repository. 7 Records were left out because those specimens originated from outside Belgium and missed coordinates.

For more information on the RBINS collection, please see Dekoninck *et al.* 2011 [9] and 2013 [10].

### The Inventory dataset

The Culicidae Inventory dataset [5] (Figure 5) was created during a cross-sectional field survey from May till October 2007 and 2008 (inventory) and from August till October 2009 and May till August 2010 (validation), using a network of $CO_2$-baited Mosquito Magnet Liberty Plus (MMLP) traps (Woodstream Corp., Lancaster, PA, USA), throughout Belgium in three key habitats (urban, agriculture and nature) [11]. Each location was sampled only once for 1 week. Twenty-nine mosquito species were identified to the species level (Figure 6).

For more information on the Inventory dataset, please see Versteirt *et al.* 2009 [12], 2011 [13], 2012 [14], 2013 [11], and 2015 [15].

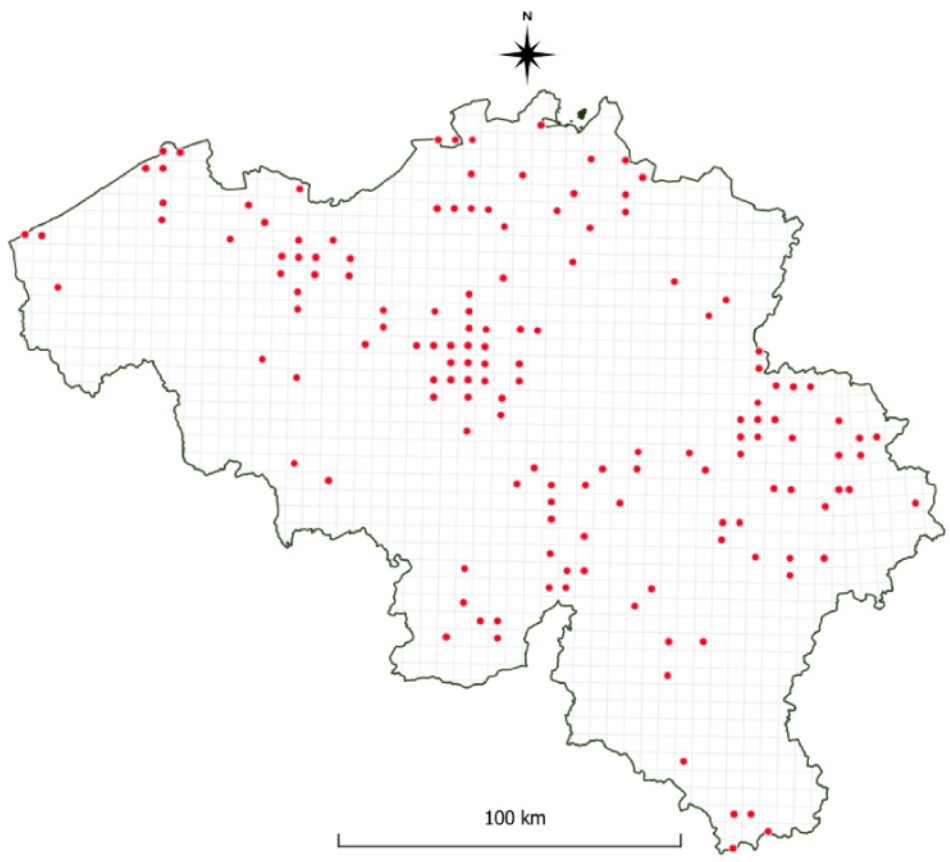

**Figure 2.** The RBINS Culicidae collection. Map of Belgium with red dots showing collection locations of Culicidae in the RBINS collection (the 'Collection dataset').

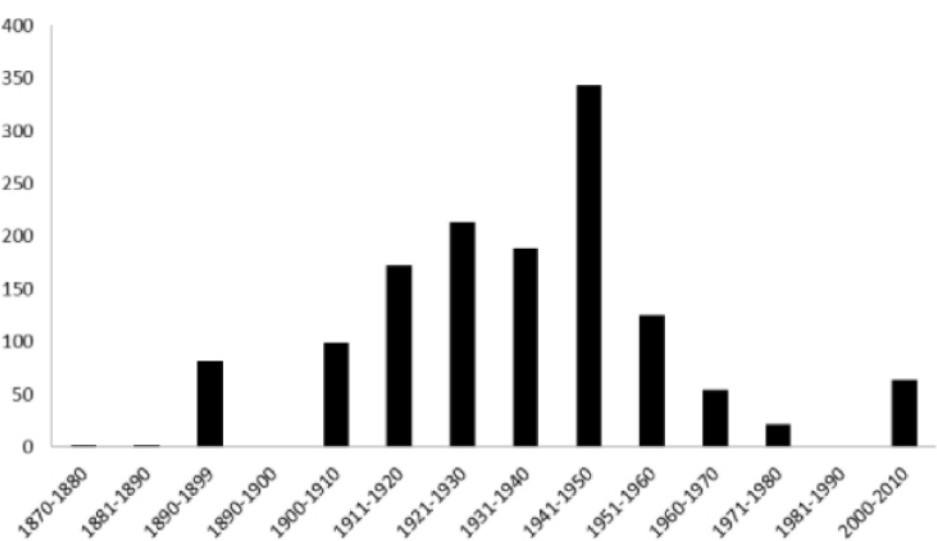

**Figure 3.** RBINS collection: distribution of voucher specimens through time.

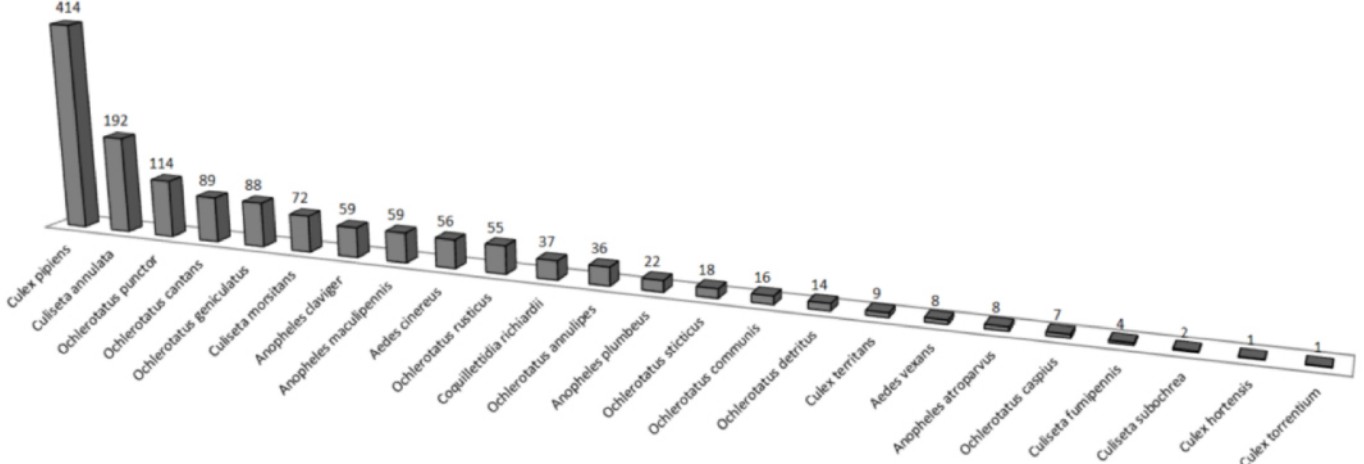

**Figure 4.** RBINS collection: species and the number of specimens available.

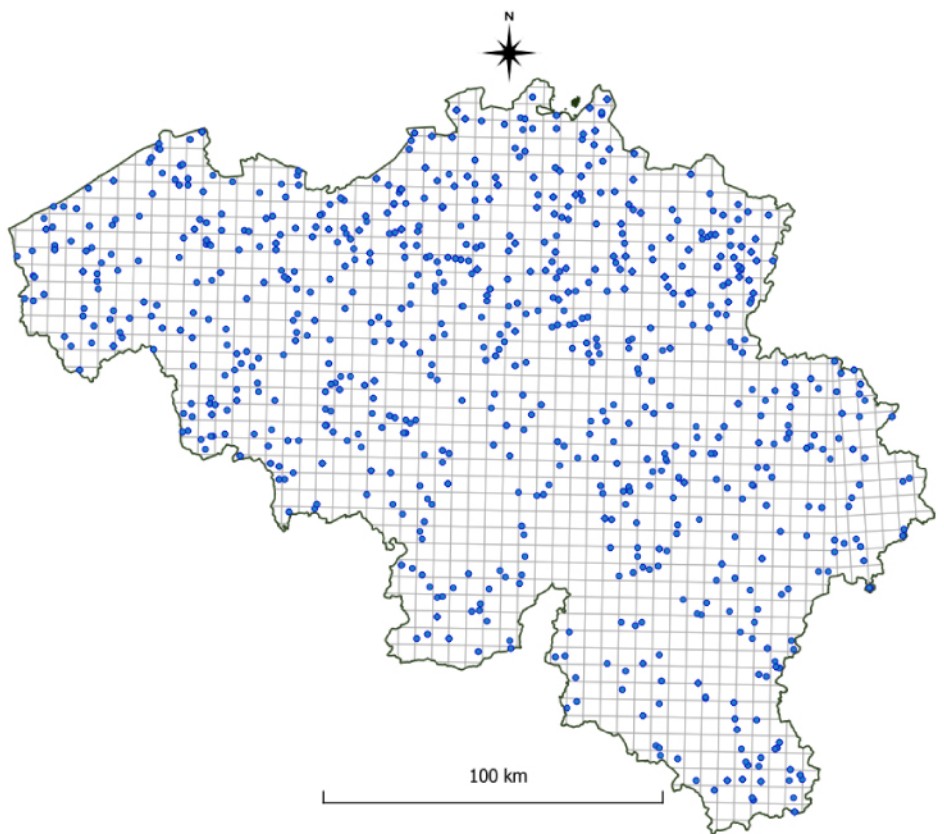

**Figure 5.** The Inventory dataset. Map of Belgium with blue dots representing Inventory collection locations (the 'Inventory dataset').

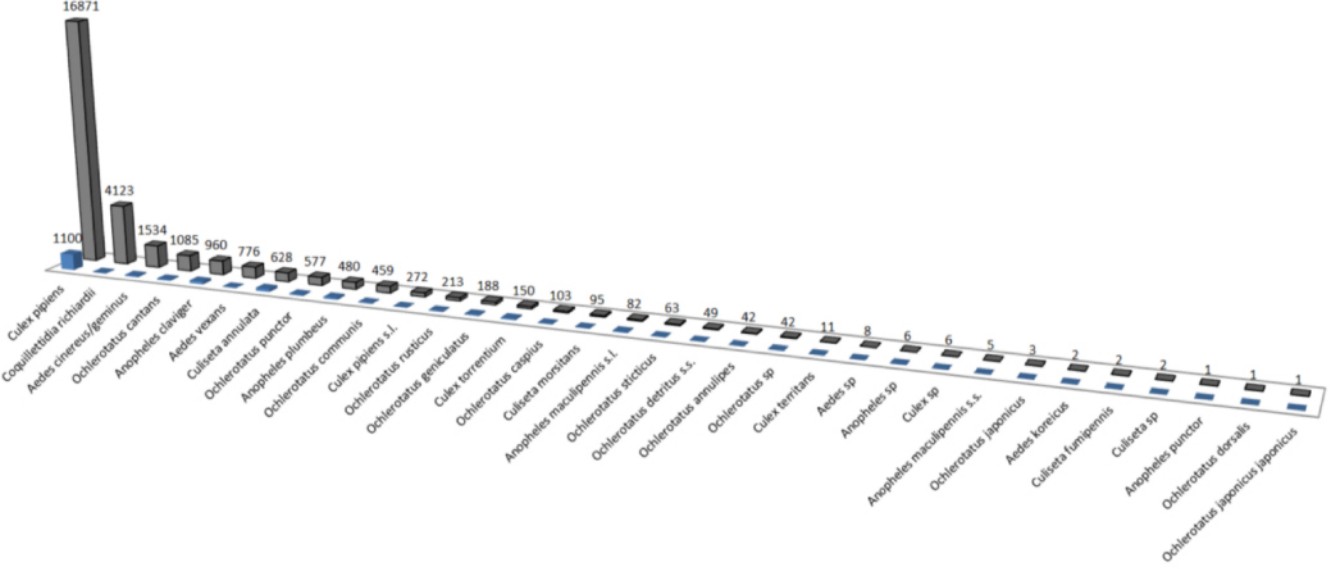

**Figure 6.** The morphological identifications and the number of individuals (grey) and the number of records (blue) in the inventory dataset.

### The Longitudinal dataset [6]

During the longitudinal study, four sites (Maasmechelen, Ruiselede, Torhout, and Natoye) were sampled thoroughly, with each study site comprising different localities (Figure 7). Each locality was sampled with a minimum of six traps: three types, two of each type, namely MMLP traps, BG Sentinel (Biogents, Regensburg, Bayern, Germany) and CDC Gravid (CDC; J.W. Hock, Gainsville, FL, USA) traps. MMLP traps were used during the inventory, BG Sentinels were used specifically to attract invasive *Aedes* species and Gravid traps were used for *Culex* species. The sites were sampled 14 times, once every 2 weeks between 20 April and 5 October 2009. In total, 36,387 adult mosquitoes ('individualCount') were collected in 651 occurrence records at the four sites (Figure 8).

#### The Maasmechelen (MA) study site

The area surveyed was an old sand quarry near Hoge Kempen national park and the Maasmechelen industrial park, where several recycling companies are located. One site, MA1, was the initial reference site, a small mixed forest fragment with birch, oak and pine next to the industrial zone. The other subsite, MA2, was on the opposite side of the road, in a narrow strip of mixed forest adjacent to a large nature reserve (heath). Land cover is largely mixed small forest, moorland, sand quarry, and a large industrial zone.

#### The Natoye (NT) study site

The population of *Aedes japonicus* at Natoye was surveyed by researchers from the Catholic University of Louvain (Université Catholique de Louvain; UCL). Two Belgian second-hand tire companies located in the village of Natoye (Namur) were surveyed. Sites were named Natoye1 (NT1) and Natoye 2 (NT2). The companies import mainly tires for trucks and heavy vehicles originating from various European countries. Tires are stacked outside, and many are exposed to rainfall, so often contain water and organic material such as decomposing

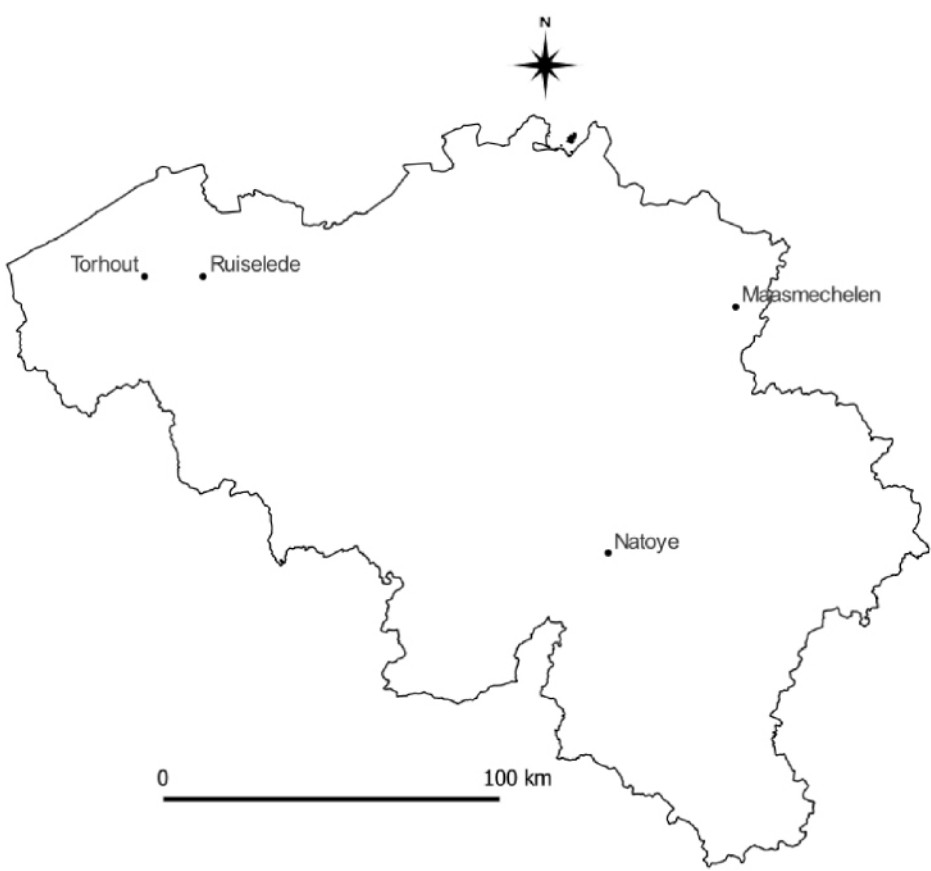

**Figure 7.** Longitudinal monitoring. Map of Belgium showing sites and subsites chosen for the Longitudinal study. MA1: Maasmechelen site 1; MA2: Maasmechelen site 2; NT1: Natoye site 1; NA2: Natoye site 2; RL01: Ruiselede site 1; TH2: Torhout site 2.

leaves. Land cover around Natoye 1 consists largely of deciduous forests, gardens and cultivated fields, while around Natoye 2 it is mostly gardens, cultivated fields and meadows.

### The Ruiselede (RL) and Torhout (TH) study sites

Mosquitos were sampled mosquitoes at two different localities in Western Flanders with the same ecoclimatic region, during one complete active season from May until October. At Torhout near Groenhove forest complex, two urban–rural landscapes were sampled (TH1 and TH2), while at Ruislede, Vorte Bossen, two natural landscape sites were sampled (RL01 and RL02).

For more information on the Longitudinal dataset, please see Damiens *et al.* 2014 [16], Dekoninck *et al.* 2011 [17], and Versteirt *et al.* 2012 [14].

### Taxonomic coverage of the three datasets

Morphological identification of Culicidae was done mainly using an electronic [8] and a paper identification key [18]. If a sample was too damaged for accurate morphological identification, sp. was used to describe the level of identification. In the period 2000–2009, substantial changes were proposed for the Aedini tribe taxonomy, which resulted in almost

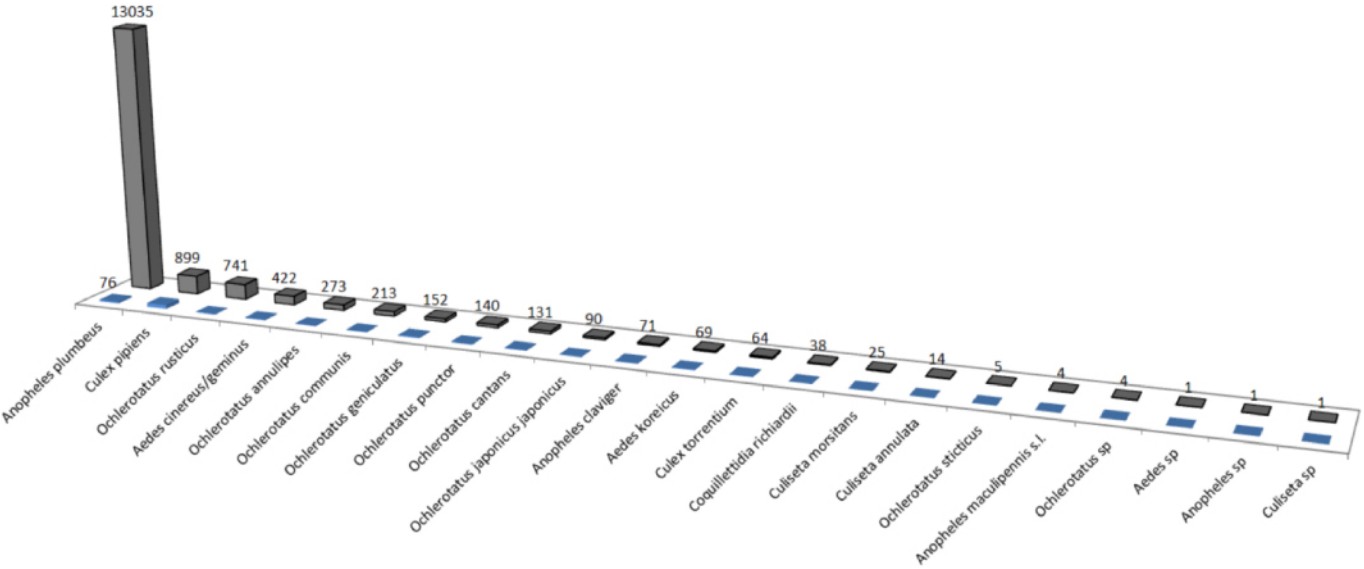

**Figure 8.** The morphological identifications and the number of individuals (grey) and the number of records (blue) in the longitudinal dataset.

tripling the number of genera in the entire Culicidae family. A recent publication [19] proposed to return to the taxonomy from before 2000, restoring a classification system useful for the operational community. This dataset was built during 2000–2009, using the available taxonomies at that time, resulting in the consequent use of the genus *Ochlerotatus* (Now *Aedes*) in the dataset.

### Geographic coverage of the three datasets

Belgium is a small country in Western Europe. To the west, its 70 km coastline fronts the North Sea; to the north lies the Netherlands; to the east, Germany, and to the south, France and Luxembourg. Biogeographically, the fauna of eastern Belgium belongs to the Central European province of the Eurasian (Palearctic) region. By contrast, the rest of the country primarily consists of an Atlantic fauna (Atlantic Biogeographical Region) (Figure 9).

Politically and geographically, the country is divided into three parts: Flanders, Wallonia and the Brussels Capital Region. In Flanders (13,522 km$^2$ with a population of about 6 million people), to the north, soils are mainly sandy to loamy. The Brussels Capital Region is a small region (162 km$^2$) entirely situated in the sandy loam area. In Wallonia (17,006 km$^2$ and about 3.5 million people), to the south, soils and habitats are more diverse, ranging from forests to rocky and calcareous grasslands on loam and chalky soils. Eastern Wallonia, near the German border, includes the Hautes Fagnes, a large area of bogs and peat.

Belgian has a temperate maritime climate that is influenced by the North Sea and the Atlantic Ocean with substantial precipitation in all seasons. Belgium has a temperate maritime climate characterised by moderate temperatures, prevailing southerly to westerly winds, abundant cloud cover and frequent precipitation. Summers are relatively cool and humid and winters relatively mild and rainy.



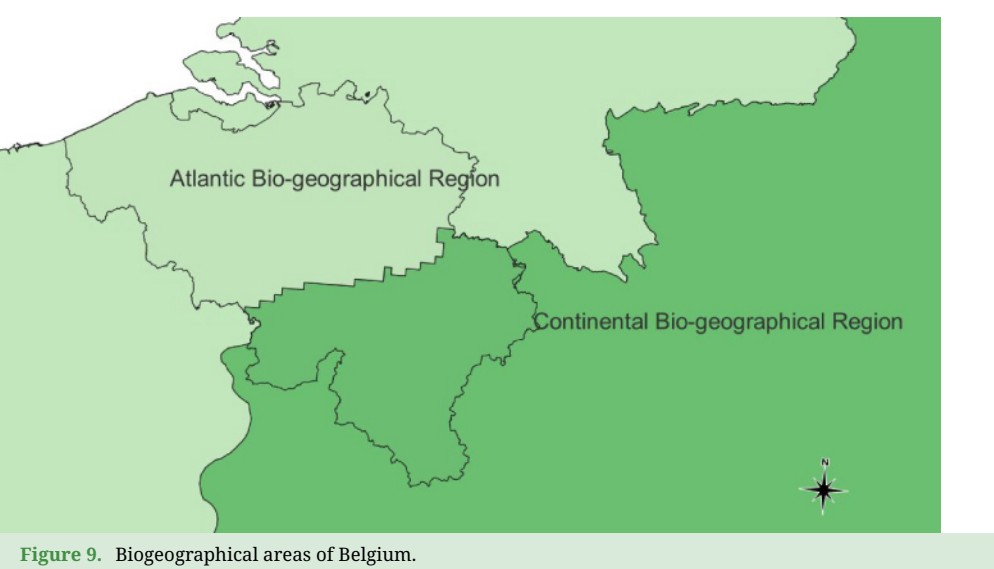

**Figure 9.** Biogeographical areas of Belgium.

### Geographical method

*Collection dataset (1878–2007).* The Universal Transverse Mercator Projection (UTM), an adaptation of the standard Mercator projection, uses a two-dimensional Cartesian coordinate system to identify locations on the surface of the Earth [20].

*The Inventory dataset (2007–2010).* The 971 selected sample sites have exact GPS coordinates. Cross-sectional field surveys were conducted during the first phase of the project to create an inventory of Culicidae. Using the NGI (National Geographic Institute) CORINE land cover (2000) map classification [21], three strata were aggregated: urban, rural and natural. In each stratum, random points were assigned, amounting to 971 selected sampling points. The number of points assigned for each Corine land cover aggregated class was proportional to its total surface in Belgium. For details of the sampling design, see Versteirt *et al.* 2011 [13] and 2013 [11].

*The Longitudinal study (2009).* All the selected sampling sites have exact GPS coordinates.
The overall distribution of the mosquito records over time is illustrated in Figure 11.

### DATA VALIDATION AND QUALITY CONTROL

To assure the quality of morphological identifications, a random sample (10%) of the identified mosquitoes collected through the MODIRISK project (Inventory and Longitudinal datasets) was re-identified by an external expert. The re-identification of the RBINS collections done during the MODIRISK project was also checked by an external expert.

Members of the *An. maculipennis* complex were further identified to species level using PCR of the ITS2 region [22].

### DATASET DESCRIPTION

Occurrence data from the MODIRISK database are extracted, standardized, and published as three separate Darwin Core archives: the Collection, the Inventory and the Longitudinal study. The main rationale behind this is that these different datasets are built for their own

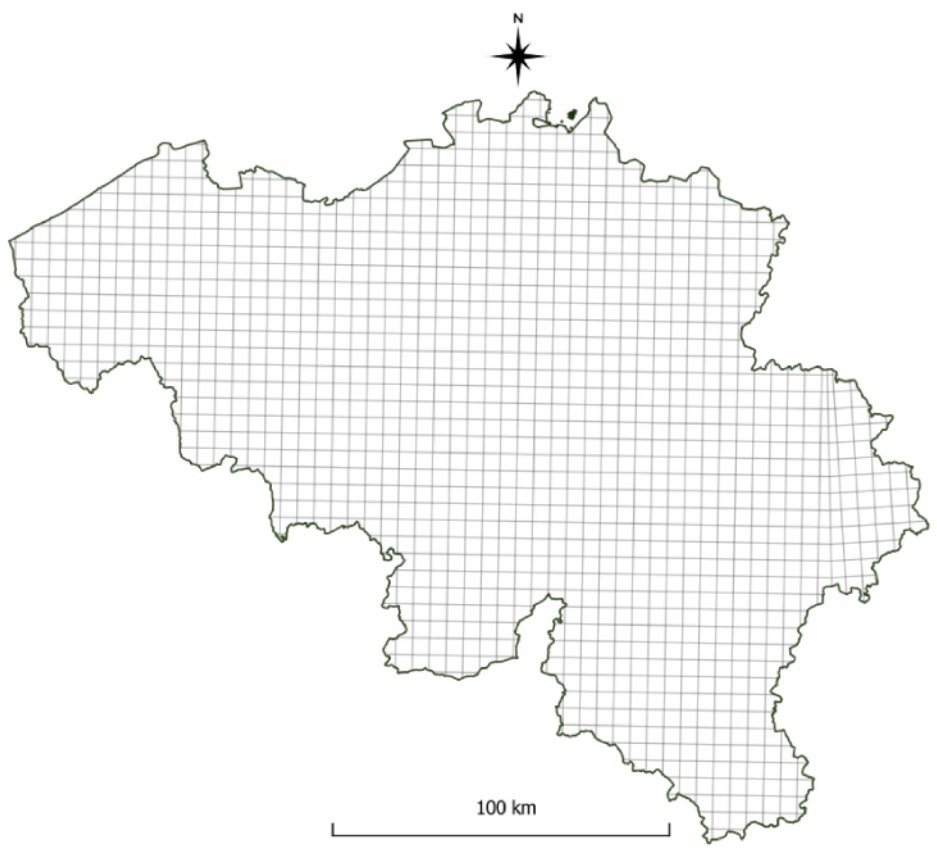

**Figure 10.** The Universal Transverse Mercator Projection 5 km grid of Belgium (Figure 10).

specific purpose and differ in sampling protocols and methods. Together these datasets represent a complete overview of the data collected during the MODIRISK project. We made the data available to GBIF to make future use, such as Species Distribution Modelling, possible.

The Darwin Core terms in the dataset at the time of publication can be found here [23].

## DARWIN CORE DATA STRUCTURE

## Collection dataset

### Occurrence core

```
Id; type; language; license; rightsHolder; accessRights; institutionCode;
collectionCode; datasetName; ownerInstitutionCode; basisOfRecord; occurrenceID;
catalogNumber; recordedBy; sex; lifeStage; preparations; disposition; eventDate;
year; month; day; countryCode; stateProvince municipality; locality;
verbatimCoordinates; verbatimLatitude; verbatimLongitude; decimalLatitude;
decimalLongitude; geodeticDatum; georeferenceRemarks; identifiedBy;
scientificNameoriginalNameUsage; kingdom; genus; specificEpithet;
nomenclaturalCode.
```

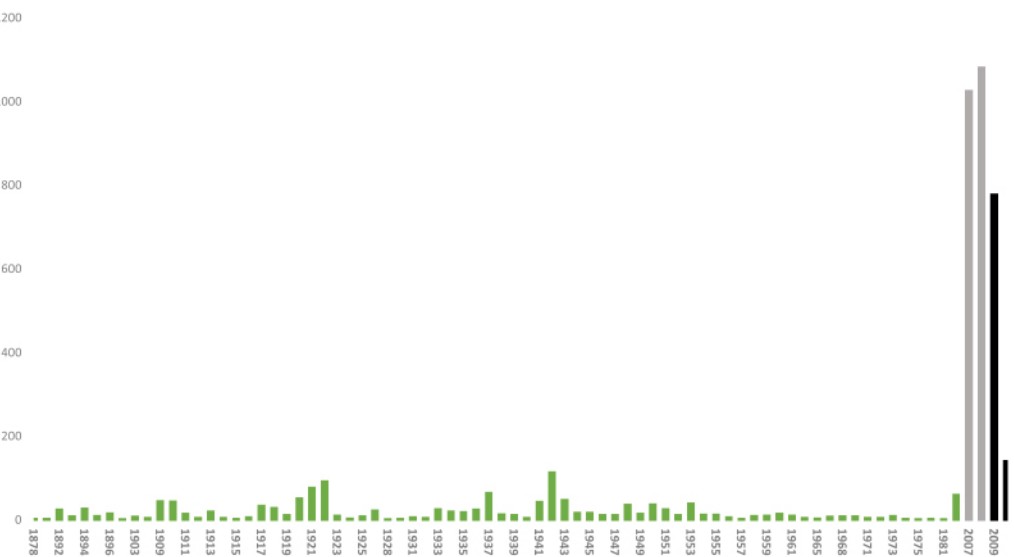

**Figure 11.** Distribution of occurrences of all three datasets through time (red: Culicidae collection; blue: Inventory; black = Longitudinal study).

- **Object name**: Darwin Core Archive MODIRISK:RBINS Diptera: Culicidae Collection
- **DOI**: https://doi.org/10.15468/3in3fb
- **Character encoding**: UTF-8
- **Format name**: Darwin Core Archive format
- **Format version**: 1.12
- **Distribution**: http://ipt.biodiversity.be/archive.do?r=modirisk-rbins-culidae-collection
- **Last Publication date of data**: 2021-03-25
- **Language**: English
- **Licences of use**: CC0 1.0 Universal (CC0 1.0) Public Domain Dedication
- **Metadata language**: English
- **Date of metadata update**: 2022-02-27
- **Hierarchy level**: Dataset

## Inventory dataset
### Occurrence core

```
id; type; license; institutionCode; basisOfRecord; dynamicProperties; occurrenceID;
individualCount; sex; samplingProtocol; eventDate; year; verbatimEventDate;
habitat; continent; country; countryCode; stateProvince; municipality; locality;
locationRemarks; decimalLatitude; decimalLongitude; geodeticDatum; identifiedBy;
scientificName; genus; specificEpithet
```

- **Object name**: Darwin Core Archive MODIRISK: Monitoring of Mosquito Vectors of Disease (inventory)
- **DOI**: https://doi.org/10.15468/4fidg2
- **Character encoding**: UTF-8
- **Format name**: Darwin Core Archive format
- **Format version**: 3.6

- **Distribution**: http://ipt.biodiversity.be/archive.do?r=modirisk-monitoring-2
- **Last Publication date of data**: 2022-02-28
- **Language**: English
- **Licences of use**: CC0 1.0 Universal (CC0 1.0) Public Domain Dedication
- **Metadata language**: English
- **Hierarchy level**: Dataset

### Longitudinal-study
#### Event core
```
id; type; language; license; rightsHolder; accessRights; datasetIDinstitutionCode;
datasetName; ownerInstitutionCode; eventID; parentEventID; samplingProtocol;
eventDate; year; month; day; verbatimEventDate; eventRemarks; countryCode;
municipality; locality; verbatimLatitude; verbatimLongitude; decimalLatitude;
decimalLongitude; geodeticDatum
```

#### Occurrence extension
```
Id; type; language; license; rightsHolder; accessRights; institutionCode;
datasetName; ownerInstitutionCode; basisOfRecord; occurrenceID; individualCount;
eventID; parentEventID; samplingProtocol; eventDate; verbatimEventDate;
eventRemarkscountryCode; municipality; locality; verbatimLatitude;
verbatimLongitude; decimalLatitude; decimalLongitude; geodeticDatum;
scientificName; kingdom; genus; specificEpithet; nomenclaturalCode
```

- **Object name**: Darwin Core Archive MODIRISK:Monitoring of Mosquito Vectors, Longitudinal study
- **DOI**: https://doi.org/10.15468/rwsozv
- **Character encoding**: UTF-8
- **Format name**: Darwin Core Archive format
- **Format version**: 1.1
- **Distribution**: http://ipt.biodiversity.be/archive.do?r=modirisk-longitudinal-culicidae-study
- **Last Publication date of data**: 2022-02-28
- **Language**: English
- **Licences of use**: CC0 1.0 Universal (CC0 1.0) Public Domain Dedication
- **Metadata language**: English
- **Hierarchy level**: Dataset

### DATA AVAILABILITY
This data paper is linked with three MODIRISK mosquito related datasets: the Collection [4], Inventory [5], and Longitudinal study datasets [6]. The database server uses Windows Server 2003 SBS R2 as the operating system, and is running IIS with PHP for site development, MS SQL Server for database development and SQL Server Mobile Tools to allow remote access from a PDA. Three types of MODIRISK forms were prepared by the MODIRISK coordinator and adapted during a group session: the field form, morphological identification form, and mosquito storage form. Based on these, relevant tables were developed by Avia-GIS [24], implemented in the database, and transferred to the web server.



The data are published under a Creative Commons CC0 waiver and we kindly ask you to notify the corresponding authors of the respective dataset if you use the data, especially for research purposes.

## EDITOR'S NOTE

This paper is part of a series of Data Release articles working with GBIF and supported by the Special Programme for Research and Training in Tropical Diseases (TDR), hosted at the World Health Organization [25].

## DECLARATIONS
## LIST OF ABBREVIATIONS

UTM: Universal Transverse Mercator.

## ETHICAL APPROVAL

Not applicable.

## CONSENT FOR PUBLICATION

Not applicable.

## COMPETING INTERESTS

The authors declare that they have no competing interests.

## FUNDING

The project 'Mosquito vectors of disease: spatial biodiversity, drivers of change, and risk' was funded by Belspo, under the Science for Sustainable Development projects SD/BD/04B and SD/BD/04D [26–28].

## AUTHORS' CONTRIBUTIONS
**Principal investigators**: Wouter Dekoninck; Wim Van Bortel; Veerle Versteirt
**Resource contact, resource creator, point of contact**: Wouter Dekoninck; Wim Van Bortel; Veerle Versteirt
**Metadata provider**: Dimitri Brosens
**Content providers**: Wouter Dekoninck; Wim Van Bortel; Veerle Versteirt
**Processors**: Dimitri Brosens.

## ACKNOWLEDGEMENTS

The authors would like to thank everybody who contributed to the creation of these datasets and paper, including the Belgian Biodiversity Platform [29], and GBIF [30]. Special thanks to the to Annemie Van Ranst, Bram Wellekens, Cleo Pandelaers, Rien Dekeyzer, Marijke Wouters, Stefan Kerkhof and Charlotte Sohier without whom the field work would not been done so smoothly. Special gratitude to Thomas Little for checking the English language.

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
