## [Reviewer Report]

Upload additional filesDRR-202202-05/form/DRR-202202-05_Data-Review-CIH.pdfReviewer name and names of any other individual's who aided in reviewer Christopher HunterDo you understand and agree to our policy of having open and named reviews, and having your review included with the published papers. (If no, please inform the editor that you cannot review this manuscript.)YesIs the language of sufficient quality?YesPlease add additional comments on language quality to clarify if needed
Are all data available and do they match the descriptions in the paper? NoAdditional CommentsThe data are available, but there are some inconsistencies between the descriptions provided in the manuscript and the data as available from GBIF. Please see the additional PDF file attached and the final comment section below.Are the data and metadata consistent with relevant minimum information or reporting standards? See GigaDB checklists for examples <a href="http://gigadb.org/site/guide" target="_blank">http://gigadb.org/site/guide</a>YesAdditional CommentsIs the data acquisition clear, complete and methodologically sound?YesAdditional CommentsIs there sufficient detail in the methods and data-processing steps to allow reproduction?YesAdditional CommentsIs there sufficient data validation and statistical analyses of data quality? YesAdditional CommentsIs the validation suitable for this type of data?YesAdditional CommentsIs there sufficient information for others to reuse this dataset or integrate it with other data?YesAdditional CommentsAny Additional Overall Comments to the AuthorThis manuscript describes the collection of 3 GBIF datasets in relation to an overarching project called MODIRISK, aimed at monitoring and predicting mosquito biodiversity changes in Belgium.
The 3 collections cover the geographic location of Belgium, over the period 1878 to 2011.

Number of GBIF datasets included in the manuscript = 3
The Collection dataset 
The Inventory dataset
The longitudinal study

Major comments:
1 - In general this manuscript is very difficult to compare to the 3 separate GBIF datasets, this is exacerbated by the inconsistency between some of the numbers quoted in the MS compared to those in the GBIF data (as outlined below). Care should be taken to consistently refer to the datasets using 1 name or unique ID for each. Where there are differences between the dataset as originally archived in the host institute and the dataset as presented in GBIF these should be highlighted and explained, or where possible, fixed in one or both of the repositories.The GBIF “known issues” information is a good starting point as it highlights those issues that arise from automated processing of the submitted dataset(s).

2 – “The Longitudinal dataset” should be checked carefully as the number and names of the sites are inconsistent within the manuscript. And the number of occurrences in the GBIF dataset is approximately half that of the number expected from reading the manuscript. (see below for details).

Minor comments:
1 - The following sentence appears to be the first mention of the 3 datasets by name and therefore should include the relevant references to the 3 later mentioned publications. “These three datasets are closely linked, but are published as three different Darwin Core Archives. The Inventory dataset: “MODIRISK: Monitoring of Mosquito Vectors of Disease (inventory)” was first published in 2013 (occurrence core) while the Collection dataset (occurrence core) and the longitudinal study (event Core) were published in 2017.”

2 - Under the heading “Temporal coverage” the authors have written “The Collection dataset (1812-2005),” which is disparate with the actual dates of that study, I believe it should be 1878-2007.

3 - On the last line of page 10 “...records at the four sites (figure 8).” but Figure 8 has 6 points highlighted on the map of Belgium?. Figure 8 also uses different site acronyms to those used in both GBIF data and in the main text of the manuscript. 

4 – The number of occurrences in “The Collection dataset” (see below for details) should be checked and corrected where appropriate. Numbers from 4 sub-collections do not add up: 135+269+241+737 = 1382 not 1381 as stated in MS. Also GBIF only contains 1374?

5 – Check and correct if appropriate the collection dates for “the Inventory study” stated in the manuscript as 2007-2008 

6 - Incorrect URL to “the Inventory dataset” IPT (manuscript page 10) http://ipt.biodiversity.be/resource?r=modirisk-longitudinal-culicidae-study, It should be https://ipt.biodiversity.be/resource?r=modirisk-monitoring-2 

7 - Figure 9: Biogeographical areas of Belgium- This is a map with some coloured regions but it is unclear what the colours represent or where the information came from, the figure legend should be improved. The text in the MS refers to Figure 9 saying there are 2 regions of different fauna, but the map includes 4 different coloured regions?

Please see the additional PDF file uploaded for more details about the above comments.RecommendationMinor Revision

---

## [Reviewer Report]

Reviewer name and names of any other individual's who aided in reviewer Maria Anice Mureb SallumDo you understand and agree to our policy of having open and named reviews, and having your review included with the published papers. (If no, please inform the editor that you cannot review this manuscript.)YesIs the language of sufficient quality?YesPlease add additional comments on language quality to clarify if needed
None. English quality is adequate.Are all data available and do they match the descriptions in the paper? YesAdditional CommentsYes, the data is available and they match the description in the manuscript. Are the data and metadata consistent with relevant minimum information or reporting standards? See GigaDB checklists for examples <a href="http://gigadb.org/site/guide" target="_blank">http://gigadb.org/site/guide</a>YesAdditional CommentsYes, the data and metadata are consistent.Is the data acquisition clear, complete and methodologically sound?YesAdditional CommentsYYes, authors collected data from museums and from field collections using district mosquito traps to ensure they would sample distinct groups. Also, they employed both transversal and longitudinal field collection data. 


Is there sufficient detail in the methods and data-processing steps to allow reproduction?NoAdditional CommentsNo, there is not enough details in the data processing to allow reproduction. Is there sufficient data validation and statistical analyses of data quality? NoAdditional CommentsIs the validation suitable for this type of data?NoAdditional CommentsIs there sufficient information for others to reuse this dataset or integrate it with other data?YesAdditional CommentsAny Additional Overall Comments to the AuthorSpecimens of some genera were not identified to species level and are grouped into genus, such as:

Aedes sp.
Anopheles sp.
Culiseta sp.
Ochlerotatus sp.
Aedes cinereus/geminus

In the figure 8 authors quoted as species despite they were not identified to species level. Authors should correct the information in accordance with the identification level. 

It is not clear if Aedes sp. (and others) is cited for one specimen only or a pool of specimens. If it is a pool, authors should use Aedes spp.

In the The Longitudinal dataset (starting in line 132), authors mentioned three sites and subsites. I do not know if they can call subsites, I would like to suggest them to use localities and sites / points. It is confusing because they mentioned three, then four, but there are five in figure 7, and from line 144 to line 160 there cite 8, likely subsites. It is important to the use the same termsto refer to localities, sites and subsites.

Figure 8 - Aedes spp., Anopheles spp. Are not species. It is not clear if the numbers above the bars are the specimen numbers or number of collections in which the species were sampled.

Taxonomic ranks
Please, not use italic fto cite [Kingdom: Animalia Class: Insecta, Orders: Diptera, Families: Culicidae Subfamilies: Anophelinae & Culicinae ]. Authors, add space after [:] [Subfamilies:Anophelinae].

In the paragraph between lines 190-197 they mentioned ants. This insect is not the focus of the study. I do not know whey the mentioned ants.

Line 199 - Are the summer moderate and the winter mild regarding air temperature or rain?

There are few typos to be checked and correct throughout the text.
RecommendationMinor Revision

---

## [Reviewer Report]

Reviewer name and names of any other individual's who aided in reviewer Abdelghafar AlkisheDo you understand and agree to our policy of having open and named reviews, and having your review included with the published papers. (If no, please inform the editor that you cannot review this manuscript.)YesIs the language of sufficient quality?YesPlease add additional comments on language quality to clarify if needed
I just found some places in this manuscript that need to be reviewed by an English speaker due to a lot of connected sentences that made me correct or put a comma to make it understandableAre all data available and do they match the descriptions in the paper? YesAdditional CommentsAre the data and metadata consistent with relevant minimum information or reporting standards? See GigaDB checklists for examples <a href="http://gigadb.org/site/guide" target="_blank">http://gigadb.org/site/guide</a>YesAdditional CommentsIs the data acquisition clear, complete and methodologically sound?YesAdditional CommentsIs there sufficient detail in the methods and data-processing steps to allow reproduction?YesAdditional CommentsI suggest adding a paragraph to mention the importance of this data in terms of future work by other scientists, especially those who use ecological niche modeling approaches!Is there sufficient data validation and statistical analyses of data quality? YesAdditional CommentsIs the validation suitable for this type of data?YesAdditional CommentsIs there sufficient information for others to reuse this dataset or integrate it with other data?YesAdditional CommentsAny Additional Overall Comments to the AuthorRecommendationAccept